# Real-Time Production and Logistics Self-Adaption Scheduling Based on Information Entropy Theory

**DOI:** 10.3390/s20164507

**Published:** 2020-08-12

**Authors:** Wenchao Yang, Wenfeng Li, Yulian Cao, Yun Luo, Lijun He

**Affiliations:** 1School of Logistics Engineering, Wuhan University of Technology, Wuhan 430063, China; yangwc6532@gmail.com (W.Y.); yunluo@whut.edu.cn (Y.L.); helj@whut.edu.cn (L.H.); 2School of Aviation, University of New South Wales, Sydney, NSW 2052, Australia; yalianjingren@126.com

**Keywords:** industrial internet of things, random job arrival time, information entropy theory, self-adaption, real-time scheduling

## Abstract

In recent years, the individualized demand of customers brings small batches and diversification of orders towards enterprises. The application of enabling technologies in the factory, such as the industrial Internet of things (IIoT) and cloud manufacturing (CMfg), enhances the ability of customer requirement automatic elicitation and the manufacturing process control. The job shop scheduling problem with a random job arrival time dramatically increases the difficulty in process management. Thus, how to collaboratively schedule the production and logistics resources in the shop floor is very challenging, and it has a fundamental and practical significance of achieving the competitiveness for an enterprise. To address this issue, the real-time model of production and logistics resources is built firstly. Then, the task entropy model is built based on the task information. Finally, the real-time self-adaption collaboration of production and logistics resources is realized. The proposed algorithm is carried out based on a practical case to evaluate its effectiveness. Experimental results show that our proposed algorithm outperforms three existing algorithms.

## 1. Introduction

With the growing demand for product customization, the characteristic of orders is developing towards diversification and small batch. Usually, the manufacturers’ goal is to minimize the completion time and energy consumption. The factory arranges the manufacturing task according to its own goal, which rarely considers customers’ requirements [1]. The scheduling problem with random orders has been becoming increasingly important for smart manufacturing. Traditionally, the manufacturing factory focuses on reducing production cost [2]. However, the goal of a smart factory involves not only minimizing manufacturing cost, but also maximizing customer satisfaction.

In addition, the manufacturers’ delivery delay will lead to reducing customer satisfaction and increasing costs. Under the cloud manufacturing (CMfg) mode, each customer wants to receive the required products from a smart factory within an expected date [3]. The manufacturers shall meet the customer’s delivery time while considering their own production capacity limitations. Traditional scheduling assumes that information on production resources and tasks is fully known. This assumption is not applicable in a customized environment because customer orders will arrive dynamically [4]. In the intelligent plant, the instantaneity demand of customers dramatically increases the difficulty of job-shop scheduling problem (JSP), i.e., random job arrival increases the complexity of scheduling problems [5].

In a new manufacturing environment, such as the Internet of things and CMfg, tasks of smart shop floor (SSF) are ordered by a cloud platform, even by customers directly [6]. When the production and logistics resources are confronted with orders with a random arrival and different due date, the traditional scheduling methods will be difficult to cope with such a new manufacturing scenario. In the smart workshop, the coordination of production and logistics resources, as well as the random arrival time of orders and different due dates, should be considered. Multi-resource collaboration and resources matching optimization are vital problems that need to be solved by the smart manufacturing system in a dynamic environment based on real-time serviceability.

Production and logistics\manufacturing resources (PLRs\MRs) collaborative scheduling is the basis for improving production efficiency and resolving the problem of insufficient production resources caused by the diversified needs of customers. However, there is little guidance about how production and logistics resources collaboration allocation affects the production decision of the remanufacturing system, especially in the real-time manufacturing system. In reality, production and processing are main stages that consume resources and energy, as well as the critical nodes with the most significant potential for intelligence. Hence, this phenomenon raises some questions. How should manufacture enterprises manage heterogeneous production and logistics resources to realize intelligence in manufacturing? Under the uncertain environment, how should the manufacturing system dynamically allocate real-time tasks?

This paper proposes a production and logistics real-time adaptive scheduling method based on information entropy to solve the above problems. The real-time scheduling method includes three parts. (1) It builds a real-time state model of production and logistics resources for dynamic production tasks based on real-time data of SSF; (2) it builds a standard entropy model of real-time tasks according to the execution status and delivery time of customized orders; (3) based on the standard entropy model of tasks and the real-time state model of resources, a real-time scheduling algorithm based on the information entropy theory (RTSIET) is constructed for improving the efficiency of producer services.

The main contributions of this study are summarized as follows:The RTSIET strategy based on adaptive coordination of smart resources can effectively deal with tasks with time constraints. It includes features that are rarely mentioned before, such as the allocate service resources according to due date.The adaptive scheduling strategy reduces production time, energy consumption, and delays through the optimization of feasible services. In addition, compared with traditional scheduling strategies, the RTSIET strategy developed in this paper can improve coordination ability among PLRs and enhance the stability of real-time scheduling.

The remaining of this paper is organized as follows. Section 2 gives a literature review. Section 3 describes the problem of real-time manufacturing resources allocation (RTMRA) with a scenario description. Section 4 presents the conceptual model and the information model. Section 5 describes the overall solution and fundamental algorithms for RTMRA. Based on a practical case, Section 6 represents and analyzes the research results of this paper. Section 7 highlights the conclusions and provides future works.

## 2. Related Work

The flexible job-shop scheduling problem (FJSP) is an outgrowth of the classic job shop scheduling problem (JSP), which has multi-function machines [7]. During the past three decades, there has been extensive development of efficient techniques for solving the FJSP in the traditional manufacturing industry [8]. Evolutionary algorithms (EAs), such as particle swarm optimization (PSO) algorithm [9] and fireworks algorithm (FWA) [10], show advantages in solving optimization problems. Some popular EAs, such as PSO, FWA, genetic algorithm, etc., which are applied to traditional FJSP, have achieved excellent results [11,12]. Although researchers have tackled the JSP with various brilliant approaches, there are limitations when dealing with practical implementation under an ever-changing modern environment where a real-time scheduling decision is required due to unpredictable systems disturbances at any second [13].

The production of a smart flexible job shop (SFJS) with a distinct due window of multi-customer increases the difficulty of job shop scheduling. Hence, traditional flexible job shop scheduling methods are difficult to adapt to new conditions. As a kind of data-driven knowledge model, the manufacturing resource allocation (MRA) model is of critical importance in the manufacturing industry, which determines the efficiency and flexibility of a shop floor and its production system [14]. Smart manufacturing resources, e.g., machines, vehicles, and work-in-progress (WIP), in SFJS have self-configuration, self-learning, and self-decision intelligence [15,16].

In the environment of industrial Internet of things (IIoT), real-time manufacturing resources allocation (RTMRA) can be further developed to make full use of the interconnection among manufacturing resources to achieve intelligent cooperation [17]. Therefore, it is timely and crucial to consider adaptive scheduling and control (i.e., RTMRA) for dynamic manufacturing environments as crucial research issues in smart production management [18]. The timely feedback shop floor information during the manufacturing execution stage leads to a significant improvement in achieving real-time production scheduling [19]. Luo et al. [20,21] proposed to integrate wired and wireless networks by also taking advantage of the automated guided vehicle (AGV) in smart factories, which increases data delivery efficiency. Zhang et al. [22] presented an overall architecture of multi-agent-based real-time production scheduling to close the loop of production planning and control. Shiue et al. [23] proposed a reinforcement learning (RL)-based RTS using the multiple dispatching rules mechanism to respond to changes in the shop floor environment. Ding et al. [15] trained a hidden Markov model (HMM)-based knowledge model from the historical data for smart manufacturing resources (SMRs) to allocate themselves autonomously for manufacturing tasks. Zhang and Wang [24,25] proposed an allocation strategy based on the game optimization model for real-time tasks. Both production resources (PRs) and logistics resources (LRs) in SSF are created as an inseparable whole, yet the majority of scheduling research focused on one of them and only took the other as a constraint condition, even without any consideration [26]. The authors of [27,28] used a discrete firefly algorithm to solve one of the most common multicriteria decision making problems. The authors of [29] proposed a framework for SSF based on a cyber-physical system and agent model of manufacturing resources. Masoud et al. [30] discussed the effect of real-time data on the efficiency of a production logistics system. Azadian et al. [31] studied the operation problem of combining production scheduling with transportation planning to improve the efficiency of operation.

Pareto front is an important concept of multi-objective optimization problems [32]. Scholars have done a lot of research on the near-complete Pareto front of problems [9,10,33]. However, the current research focuses on the implementation of real-time production and logistics scheduling, and lacks research on scheduling results [34]. Furthermore, the above methods and technologies ignore the information properties of real-time tasks. Making full use of the information properties of resources and tasks in SSF is the premise and foundation of realizing intelligent manufacturing.

## 3. Problem Description and Mathematical Model

### 3.1. Problem Description

As shown in Figure 1, there are three main roles, i.e., customers, CMfg platform, and SSF in this scenario. As service demanders, customers submit order requirements to the CMfg platform. The CMfg platform decomposes orders and gets real-time jobs (i.e., product components). An SSF receives real-time manufacturing jobs, and further decomposes jobs and gets real-time tasks (i.e., manufacturing tasks), according to the real-time orders information, real-time status information of resources in SSF, and product process requirements. Then, SSF obtains logistics tasks according to the production and logistics cooperative strategy. As service providers, production and logistics equipment receive allocated manufacturing and logistics tasks and execute these tasks according to task schedules. Finally, the completed products are delivered to the service demander through logistics from the selected service provider.

The real-time allocation problem is the matching problem between manufacturing resources (including PRs and LRs) and manufacturing tasks with specific process requirements according to their status [14]. Different from the traditional scheduling problem, the RTMSA problem of multi-resource collaboration considers the heterogeneity and dynamics of SMRs, as well as the utility efficiency and sustainability of environmental impact in the manufacturing system operation stage. In addition, the influence of customer behavior on satisfaction degree is also considered.

The RTMRA can be stated as follows. Given a set of jobs $job_{set}=\left\{job^{k}|k=1,2,\cdot \cdot \cdot ,K\right\}$, a set of AGVs $A=\left\{a_{i}|i=1,2,\cdot \cdot \cdot ,I\right\}$ and a set of machines $M=\left\{m_{j}|j=1,2,\cdot \cdot \cdot ,J\right\}$. Each job has two time attributes, i.e., arrival time and due date. In addition, if a job is completed after the due date, the tardiness penalty cost is generated. The aim of RTMRA is to provide an adaptive task allocation strategy so that multiple objectives are optimized simultaneously. Assumptions are given as follows [5,35,36]:Jobs arrive randomly, and jobs have a different due date.Each operation may be executed on a set of alternative machines.The arrival time and due date of a job is not known until the job arrives.Each machine can perform only one ordinary job processing at a time.Transportation time of AGVs is considered.A task, once taken up for processing on a machine, should be completed before another task is taken.

### 3.2. Mathematical Model

To facilitate reading and understanding, Table 1 lists the mathematical symbols used in this article.

In the SSF, reducing the average task delay time of all tasks is a key scheduling optimization objective [3]. The mathematical model for RTMRA can be defined as follows. In this model, the studied objectives include makespan, total energy consumption, and mean tardiness.
(1)$$F=min\left\{f_{1},f_{2},f_{3}\right\}$$
(2)$$f_{1}=max\left\{C^{k}\right\}$$
(3)$$f_{2}=\sum_{k=1}^{d}\sum_{j=1}^{m}p_{nj}^{k}+pp_{ni}^{k}\times \sum_{k=1}^{d}\sum_{j=1}^{v}p_{ni}^{k}+\sum_{j=1}^{m}I_{j}\times p_{j}^{I}$$
(4)$$f_{3}=\frac{1}{d}\times \sum_{k=1}^{d}L^{k}\;$$

S.t.
(5)$$L^{k}=max\left(0,C^{k}-d^{k}\right)$$
(6)i={1,2,⋯,I}
(7)j={1,2,⋯,J}
(8)n={1,2,⋯,N}

In this mathematical model, $f_{1}$ denotes the makespan; $f_{2}$ denotes the total energy consumption, which includes energy consumption of machine and energy consumption of AGVs; $f_{3}$ denotes the mean tardiness of jobs.

## 4. Model Description in the Smart Shop Floor

### 4.1. Conceptual Model

The objects of data acquisition are manufacturing resources in smart factories, such as machines, AGVs, WIP, etc. Under the IIoT environment, the internal producing department can obtain real-time data in time by industrial bus, wireless sensor network, RFID reader and camera, etc. The external can obtain real-time orders by industrial cloud platform, ERP\MES, and other upper-layer applications. The dynamic characteristics of workshop resource status and order arrival time require the smart workshop resource model to differ from the traditional one [37]. The production resource service model of smart workshops should not only build its static serviceability, but also have the function of constructing a real-time service capability based on its own real-time state and task requirements.

**Definition** **1—Smart Work-in-progress (SWIP).***It refers to goods in process with a passive recognition ability in physical space. SWIP can be perceived by manufacturing resources (e.g., manufacturing equipment, processing equipment, and people) in the manufacturing environment and read, related requirements (e.g., production process, emergency grade, and deadline) of SWIP. Manufacturing resources are dynamically adjusted in the manufacturing process to coordinate the completion of production tasks*.

**Definition** **2—Smart Manufacturing Resources (SMRs).***It refers to the production process based on WIP that complete the relevant handling, processing (assembly), and quality inspection and other related resources, including production resources, logistics resources, and people with wearable devices that are capable of sensing, communication interaction, learning, execution, self-control, etc., in physical space. After establishing the business association, smart manufacturing resources and SWIP jointly complete the manufacturing task in the form of cooperation/competition with the goal of the lowest manufacturing cost, the highest manufacturing efficiency, and the lowest energy consumption*.

The smart modeling matrix set of PRs includes two parts: The attribute of resources and real-time status in the environment of IIoT. The real-time status includes dynamic queue, service load, and service process status, etc. Hence, the real-time perception of the state of key manufacturing resources in smart workshops is the basis of constructing the smart model [38]. The purpose of introducing SWIP and SMRS into the self-adaption scheduling process is to formalize product requirements, resource capabilities, attributes, and constraints.

In order to manage the real-time state data of key resources more effectively, the real-time state model of PRs (e.g., machines and numerical control machining centers) and LRs (e.g., AGVs) are constructed as follows:

At time t, the set of service types of $m_{j}$ can be described as $S_{j}^{t}=\left\{s_{j}^{\alpha }|\alpha =1,2,\cdots ,\theta \right\}$, where θ is the number of service types that $m_{j}\;$can be provided, and $s_{j}^{\alpha }$ is one of them. Meanwhile, the set of service types of $m_{j}$ can be described as $S_{i}^{t}=\left\{s_{i}^{\tau }|\tau =1,2,\cdots ,\gamma \right\}$, where γ is the number of service types that $a_{i}\;$can be provided, and $s_{i}^{\tau }$ is one of them.

The real-time status attribute of production equipment has six characteristics, including equipment number, service option, manufacturing energy consumption, idle energy consumption, and manufacturing time.
(9)$${\bar{m}}_{j}^{t}=\left(m_{j},S_{j}^{t},ep_{nj}^{k},e_{j}^{I},p_{nj}^{k},sq_{j}\right)$$
where $S_{j}^{t}$ denotes the type of service that the machine can provide, $ep_{nj}^{k}$ denotes the processing energy consumption of the machine tool for the current task, $e_{j}^{I}$ denotes the idle energy consumption of the machine tool, $p_{nj}^{k}$ denotes the service time of the machine tool for the current task,$\;sq_{j}$ denotes the service queue of $m_{j}$.

The real-time status attribute of logistics equipment is defined as seven characteristics, including equipment number, service options, location of the handling equipment, handling energy consumption, standby energy consumption, and handling time.
(10)$${\bar{a}}_{i}^{t}=\left(a_{i},S_{i}^{t},L_{i},ep_{ni}^{k},e_{i}^{I},p_{ni}^{k},sq_{i}\right)$$
where $S_{i}^{t}$ denotes the type of service provided by the handling equipment $a_{i}$, $L_{i}$ denotes the energy consumption of the handling equipment for the task, $ep_{ni}^{k}$ denotes the location of the handling equipment, $e_{i}^{I}$ denotes the idle energy consumption, $p_{ni}^{k}$ denotes the service time of the handling equipment for the current task, $sq_{i}$ denotes the service queue of $a_{i}$.

**Definition** **3—Real-time Tasks (RTs).***Generally, in the field of manufacturing, there are two types of tasks, i.e., simple tasks and complex tasks. A simple task is a basic task that can be completed independently by a single service resource. It is a definite step of a complex task. Simple tasks have positive input conditions and output results. In addition, it also contains explicit attribute features, such as task arrival and end time, resource capability demand, and task execution time, etc. In this paper, RTs refer to complex tasks. It contains two simple tasks, such as production task of*$tk_{n}^{k}$*and logistics task of*$tk_{n}^{k}$.

The production and logistics collaborative manufacturing scenario in SSF, a manufacturing task includes a production task and a logistic task [39]. In this study, the production task and logistics task of a manufacturing task are packaged and released in groups. We assume that task $tk_{n}^{k}$ will be performed by $m_{j}$ and $a_{i}$. Hence, the *input* set of $tk_{n}^{k}$ is denoted by (11).
(11)$$Input=\left(tk_{n\;}^{k},\;{\bar{m}}_{j}^{t},{\bar{a}}_{i}^{t}\right)$$
where ${\bar{m}}_{j}^{t}$ is the status of the machine $m_{j}$ which will be performing the production task of $tk_{n\;}^{k}$, ${\bar{a}}_{i}^{t}$ is the status of the AGV $a_{i}$ which will be performing the logistics task of $tk_{n\;}^{k}$. It describes the serviceability of the required before $tk_{n}^{k}$ executes at time t, including the ability of processing resources and the ability of logistics resources.

When the task is completed at time $\overset{ˇ}{t}$, the *output* set of the $tk_{n}^{k}$ is denoted by (12).
(12)$$Output=\left(tk_{n+1}^{k},{\bar{m}}_{j}^{\overset{ˇ}{t}},{\bar{a}}_{i}^{\overset{ˇ}{t}}\right)$$

It describes the status of service resources when $tk_{n}^{k}$ has been executed at time $\overset{ˇ}{t}$.

### 4.2. Real-Time Information Model of Tasks for Multi-Customer

Entropy is defined as the product of information generated by an event and the probability of the event [26]. In the scenario described in Section 3, this paper focuses on orders with different arrival times and due dates. In a real-time distributed system, the remaining execution time and deadline are some fundamental attributes of real-time tasks that elucidate the activities of the manufacturing system [35].

In the intelligent workshop layer, we can easily obtain the real-time process of jobs and the real-time status of PLs based on the conceptual model. We assume that the release time of $tk_{n\;}^{k}$ is time t, $\;tk_{n\;}^{k}\in job^{k}$. The predicted mean remaining processing time of $job^{k}$ is denoted by (13).
(13)$$rp\_tk_{n}^{k}=\overset{I}{\underset{i=n}{\sum }}\overset{r}{\underset{c=1}{\sum }}\frac{p_{nj}^{k}}{r}$$
where l is the total process number of $job^{k}$; r is the number of optional machines in each process (task) of $job^{k}$.

We assume that task $tk_{n-1}^{k}$ is processed on machine $m_{\hat{j}}$, and task $tk_{n}^{k}$ will be processed on machine $m_{j}$. The distance between machine $m_{\hat{j}}$ and machine $m_{j}$ is denoted by $Dist\left(m_{\hat{j}},m_{j}\right)$. The predicted mean remaining delivery distance of $job^{k}$ is denoted by (14).
(14)$$d_{n}^{k}=\frac{1}{J}\overset{J}{\underset{j=1}{\sum }}Dist\left(m_{\hat{j}},m_{j}\right)+\frac{\left(l-n-1\right)}{J^{2}}\overset{J}{\underset{\overset{ˇ}{j}=1}{\sum }}\overset{J}{\underset{j=1}{\sum }}Dist\left(m_{\overset{ˇ}{j}},m_{j}\right)$$
where $\hat{j},j,\overset{ˇ}{j}\in \left[1,m\right]$, $\frac{1}{J}\overset{J}{\underset{j=1}{\sum }}Dist\left(m_{\hat{j}},m_{j}\right)$ is the predicted mean remaining delivery time of $tk_{n}^{k}$, $\frac{\left(l-n-1\right)}{J^{2}}\overset{J}{\underset{\hat{j}=1}{\sum }}\overset{J}{\underset{j=1}{\sum }}Dist\left(m_{\overset{ˇ}{j}},m_{j}\right)$ is the predicted mean remaining delivery time from $tk_{n+1}^{k}$ to $tk_{J}^{k}$.

The predicted mean remaining service time of $job^{k}$ is denoted by (15).
(15)$$rpl\_tk_{n}^{k}=rp\_tk_{n}^{k}+rl\_tk_{n}^{k}$$
where $rl\_tk_{n}^{k}$ represents the predicted mean remaining delivery time of $job^{k}$, i.e., $rl\_tk_{n}^{k}=\frac{d_{n}^{k}}{\bar{v}}$.

Due to the fact that each task has its due date, at time *t*, the remaining completion time of $tk_{n}^{k}$ is denoted by (16).
(16)$$rc\_tk_{n}^{k}=d^{k}-t$$

Subject to the constraints in Equations (13) to (16), we apply the information-theoretic concepts to define the following parameters [40,41,42]:

The urgency of task $U\left(tk_{n}^{k}\right)$ is the probability of execution of the task by the ratio between the predicted mean remaining service time ($rpl\_tk_{n}^{k}$) and the remaining completion time ($rc\_tk_{n}^{k}$) of the task. At time *t*, the urgency of $tk_{n}^{k}$ is denoted by (17).
(17)$$U\left(tk_{n}^{k}\right)=\frac{rpl\_tk_{n}^{k}}{rc\_tk_{n}^{k}}$$

The normalized urgency of a task is the probability of a task normalized by the sum of all the tasks’ urgency. We assume that the total number of the tasks in a task-pool at time t is x. The tasks in a task-pool can be described as $tk_{n}^{kb}$, where b∈[1,x]. The normalized urgency of a task in a task-pool at time t is denoted by (18).
(18)$$NU\left(tk_{n}^{kb}\right)=\frac{U\left(tk_{n}^{kb}\right)}{\sum_{b=1}^{x}U\left(tk_{n}^{kb}\right)}$$
where $NU\left(tk_{n}^{k}\right)=NU\left(tk_{n}^{kb}\right)$ at time t.

The urgency of the task is a vital attribute under an uncertain scheduling environment. We define this attribute as a standard entropy of $tk_{n}^{k}$, which is formulated as follows:(19)$$NE\left(tk_{n}^{k}\right)=-\log_{2}NU\left(tk_{n}^{k}\right)$$

## 5. The Proposed Method

Two characteristics of customers’ dynamic demands are considered in this paper, namely the arrival time and due date of orders. In order to handle dynamic customer demand, a PLRs adaptive scheduling method is proposed. Figure 2 displays the flowchart of the PLRs adaptive scheduling method. The real-time scheduling method consists of two key parts, i.e., task trigger rules and entropy-based scheduling strategy.

### 5.1. Task Trigger Rules

Task triggering consists of three steps. The first one is at the beginning of execution when jobs are released from the cloud to the job pool in the SSF. The SSF should divide jobs into tasks according to the production process. Then, the first task of the job is put into the task-pool, a set of tasks in the task-pool denoted as $Tk^{t}$. For example, $tk_{1}^{k}$ is put into the task-pool at the beginning. Tasks in the task-pool will trigger the entropy-based scheduling strategy. Then, in the middle of execution, when $tk_{n}^{k}$ is successfully allocated, $tk_{n}^{k}$ will be deleted and $tk_{n+1}^{k}$ will be added to the task-pool. Finally, the above steps are repeated until the last task of the job is processed.

### 5.2. Entropy-Based Scheduling Strategy

When the scheduling policy is triggered, the scheduling center can query optional machines and optional AGVs according to the task type. We will get the optional production resources set $\bar{M}$ and optional logistics resources set $\bar{A}$. The set of the status of $\bar{M}$ and the set of the status of $\bar{A}$ can be denoted by ${\bar{M}}^{t}$ and ${\bar{A}}^{t}$. In this paper, all the machines are NC machining centers, all of them have multiple kinds of capabilities, and all of the AGVs are the same type of equipment (it is noteworthy that all AGVs have the same speed, power, and types of service). Therefore, ${\bar{M}}^{t}=\left\{{\bar{m}}_{j}^{t}|j=1,2,\cdots J\right\}$ and ${\bar{A}}^{t}=\left\{{\bar{a}}_{i}^{t}|i=1,2,\cdots I\right\}$.

In that way, $G_{U}$ is the service groups set to meet the service requirements of tasks in a task-pool at time t, denoted by (20).
(20)$$G_{U}=\left\{\left(a_{i},m_{j}\right)|a_{i}\in \bar{V},m_{j}\in \bar{M}\right\}$$

Assume that $g_{y}$ is a one service group of the $G_{U}$, $G_{U}=\left\{g_{y}|y=1,2,\ldots ,U\right\}\;$and $g_{y}=\left(a_{i},m_{j}\right)$, where U=I×J. Suppose that the $tk_{n-1}^{k}$ is processed on machine $m_{\hat{j}}$, $tk_{n}^{k}$ will be processed on machine $m_{j}$. Suppose that AGV $a_{i}$ provides logistics services to $tk_{n}^{k}$, $L_{i}$ is the location of $v_{i}$ when it starts to execute the logistics task of $tk_{n}^{k}$. The distance between $L_{i}$ and machine $m_{\hat{j}}$ is denoted by $Dist\left(m_{\hat{j}},L_{i}\right)$. The pick-up time cost of AGV for $tk_{n}^{k}$ is denoted by $p_{ni\hat{j}}^{k}=\frac{Dist\left(m_{\hat{j}},L_{i}\right)}{\bar{v}}$, the delivery time cost of AGV for $tk_{n}^{k}$ is denoted by $d_{nij}^{k}=\frac{Dist\left(m_{\hat{j}},m_{j}\right)}{\bar{v}}$.

Fabricating costs of WIP, include raw material cost, time cost, and production energy consumption. The cost of raw material is the inherent cost of manufacturing, and it will not be changed by scheduling. The time cost includes manufacturing time and handling time. The manufacturing time depends on the serviceability of the equipment arranged for the process production, which will change due to the different production scheduling. The handling time depends on the service capacity of equipment (all AGVs have the same speed and energy consumption, so it is only the difference of pick-up time\delivery time caused by the different location of AGV), and the position of the working procedure before and after the work in WIP, which will change due to the different scheduling. Hence, the total service time of service group gy is denoted by (21).
(21)Ty=pnj^k+pnij^k+dnijk+max(bpnik−fpnj^k,0)+max(bpnjk−fpnik,0)
where max(bpnik−fpnj^k,0) is the waiting time of AGV i to pick up the WIP (tknk), max(bpnjk−fpnik,0) is the waiting time of the WIP (tknk) to be executed.

Suppose that $tk_{\overset{´}{n}}^{\overset{´}{k}}$ is processed before $tk_{n}^{k}$ on machine $m_{j}$. The total energy consumption of service groups $g_{y}$ is denoted by (22).
(22)$$W_{y}=bp_{ni}^{k}\times \left(p_{ni\hat{j}}^{k}+d_{nij}^{k}\right)+p_{nj}^{k}\times ep_{nj}^{k}+max\left(bp_{ni}^{k}-fp_{\overset{´}{nj}}^{\overset{´}{k}},0\right)\times e_{j}^{I}$$

Compared with global scheduling, the advantage of real-time scheduling is that it can deal with high-frequency disturbances, but its short-sightedness makes it difficult to obtain the global optimal solution. Accordingly, it is very hard to control the completion time of each job. The weight method can get better results through repeated simulation. In the real world, the situation is unpredictable. In high-disturbance real-time scheduling, it is very difficult to obtain excellent scheduling results with the fixed weight method.

SSF is characterized by high disturbance, i.e., real-time tasks. Compared with single scheduling rules, combined scheduling rules can improve the production capacity of the workshop.

This paper proposes an adaptive weight method based on information entropy. Therefore, we can build the function $f_{0}$ as the real-time evaluation function.
(23)$$f_{0}=\omega_{e}\times T_{normalization}+\left(1-\omega_{e}\right)\times W_{normalization}$$

S.t.
(24)$$\omega_{e}=1-NE\left(tk_{n}^{k}\right)$$
(25)$$T_{normalization}=\frac{T_{y}-T_{min}}{T_{max}-T_{min}}$$
(26)$$W_{normalization}=\frac{W_{y}-W_{min}}{W_{max}-W_{min}}$$
where $T_{min}$ is the minimum service time of the optional composite service, $T_{max}$ is the maximum service time of the optional composite service, $W_{min}$ is the minimum energy consumption of the optional composite service, $W_{max}$ is the maximum energy consumption of the optional composite service.

The goal of our research is to schedule the tasks in the task-pool, at a given point in time. We use $f_{0}$ to judge the service quality of service groups, and select the service group with the lowest total cost. According to the pairing result of the service groups and the tasks in the task-pool, all tasks are assigned to the optimal machines and AGVs.

In the IIoT environment, tasks may be located at different geographical locations, and the service groups supporting these tasks are complicated and heterogeneous. Without loss of generality, we use the symbol $r_{ny}^{k}$ to denote the allocation relationship between the task and service group $g_{y}$. After mapping all the tasks in the task-pool to service groups, the result of the real-time schedule can be written as:(27)$$r_{ny}^{k}:\;tk_{n}^{k}\underset{allocation}{\to }g_{y}$$
where $Tk^{t}$ is a set of tasks in the task-pool at time t, $tk_{n}^{k}\in Tk^{t}$.

It uses information entropy to balance the preference between service time and energy consumption for task allocation. It can effectively avoid the disadvantages of fixed weight method. Moreover, the proposed method can effectively control the order completion time in real-time scheduling. Based on previous studies, a real-time scheduling algorithm based on the information entropy theory (RTSIET) is proposed. RTSIET is briefly described in Algorithm 1.
**Algorithm 1** Real-time scheduling algorithm based on the information entropy theoryInput: $Tk^{t}$, t, $d^{k}$, ${\bar{M}}^{t}$,${\bar{A}}^{t}$
Output: $r_{ny}^{k}:\;tk_{n}^{k}\underset{allocation}{\to }g_{y}$
**While** (*taskpool* ==! null) **do****for**$tk_{n}^{k}$ in taskpoolCompute the standard entropy for each task in *taskpool*as formula (19)    **for**
$g_{y}$ in $G_{k}$
**do**     Compute the service quality of each group and choose     the best one using (23)   **end for**
**end for****end while**

## 6. Case Study

To conduct the experiment evaluation, we adopt a practical case from a medium robot manufacturing company located in Wuhan, China. There are multiple NC Machining Workshops in Wuhan. The rapidly developed market of online shopping in China caused the increased demand of personalized products such as robots. Companies have to offer customized services to suit the needs of customers.

In this section, a demonstrative case from a robot manufacturing enterprise of Wuhan has demonstrated the feasibility of the proposed model for RTMSA. Flexible production and logistics resources with randomly arrival jobs are considered.

### 6.1. Case Description

As shown in Figure 3, it is a layout of Hub workshop, which is a workshop of a robot manufacturer. There are six machines, one automated warehouse and some AGVs are involved in the shop-floor.

The distance among the warehouse and machines are shown in Table 2.

In the SSF as described in this section, one hub is a job. Each job can be completed through a specific sequence of production processes. One production process type refers to a task type. There are five types of tasks in the manufacturing process: Cutting (CT), turning (TU), grinding (GR), drilling (DR), and tapping (TA). As a service provider, there are six different processing equipment (CNC machines). Each machine can provide five different types of services. Each machine can execute manufacturing tasks with different service time expectations. The service time and power of the machines are shown in Table 3.

In dynamic job shops, the distribution of jobs arrival process closely follows a Poisson distribution. Hence, the time between job arrivals closely follows an exponential distribution [5]. According to the historical data of customer’s orders, the probability distribution of order arrival is analyzed. The Poisson distribution is used to describe the random arrival orders in the SSF. Suppose 15 jobs randomly arrive in 30 min. The job arrivals rate λ is equal to 0.5, respectively. Each job contains two attributes of time: Arrival time and due date, denoted by $\bar{job^{k}}=\left(a^{k},d^{k}\right)$. The relationship between the order arrival time and due date is denoted by $d^{k}=a^{k}+\bar{c}$ [5]. According to the production data of the enterprise in three months, the average operation time of the jobs is 2000 s in the hub workshop, therefore, $\bar{c}=2000\;s$.

The idle energy consumption of the machine is shown in Table 4.

The idle energy consumption of AGV is 0 because the communication cost is not considered in this paper. The speed and handling power of AGVs are shown in Table 5.

The proposed method has been performed in python language on a computer with an Intel I7 4710MQ CPU 2.5 GHz processor and 8.00 GB memory.

### 6.2. Results of the Experiments

In order to illustrate the potential of the proposed method for the multi-objective dynamic JSP, it is compared with the self-adaptive collaboration method (SCM) [33] and some common dispatching rules. A combination decision model of scheduling rules such as the longest processing time (LPT) dispatching rule, the shortest processing time (SPT) dispatching rule, and the first in first out (FIFO) dispatching rule [5].

Paper [33] views each resource (machine and AGV) in the job shop as an active entity to request the production tasks. The processing and logistics tasks will be allocated to the optimal resources according to their real-time status by using the weight method. This method only considers the real-time status of resources but does not fully consider the value engineering information such as random arrival jobs. Therefore, sometimes the resources allocation may not be suitable, thereby reducing the efficiency of the production system. Compared with the above research, which stands at the viewpoint of a real-time job shop multi-resource self-organization in the production execution stage, our model stands at the viewpoint of a real-time job shop multi-resource self-adaption for autonomous allocation of SPLA in both production execution and customer requirements stage.

According to the case study, a series of experiments are conducted in order to evaluate the contribution of the adaptive weight mechanism based on the entropy to the performance method. Table 6 contains the average and standard deviation values for makespan, energy consumption, and mean tardiness obtained across 200 runs on each instance with different numbers of AGVs. With the increase of the number of AGVs, the indexes of all methods are improved. It can be easily found from Table 6 that the proposed method has the potential to achieve the optimal solutions. It also can be found in that the mean and variance of makespan, total energy consumption, and mean tardiness of the proposed method are best among all the methods. Furthermore, it can also be seen that the SPT performs better than the SCM, and the SCM performs better than the LPT. This is mainly because compared with the SCM model, FIFO + SPT has a powerful predilection of time minimization, which can handle complex relationships between more objectives, thus resulting in a relatively better performance. However, the powerful predilection of time maximum of FIFO + LPT is hard to deal with the interaction between multiple objectives, thus resulting in the worst performance. These results illustrate that the proposed method performs better among all the methods, and it can improve the performance of the scheduling system by keeping the schedule stability and the schedule efficiency simultaneously when the jobs arrival at random.

According to the above analysis, we can observe that FIFO + LPT performs the worst. Therefore, we will not cover it in this study. Figure 4 intuitively shows the three-dimensional Pareto fronts obtained by three algorithms (i.e., RTSIET, FIFO + SPT, and SCM) with five AGVs in solving the problem. It is clear that the non-dominated solutions obtained by RTSIET get closer to the coordinate origin. In most cases, the solutions of SCM and FIFO + SPT are scattered, which means that SMC and FIFO + SPT are challenging to obtain better results stably. It implies the superiority of RTSIET compared with other algorithms in solving the proposed problem.

In order to verify the universality and effectiveness of the proposed method, we have also done a simulation verification on other production workshops (e.g., bearing support workshop and assembly workshop). Compared with other methods, the stability and effectiveness of the proposed method are still the best. Since in the high disturbance production environment, the adaptive scheduling strategy based on the information entropy can well balance time and energy consumption. The balance characteristics of the proposed method has extraordinary significance for intelligent workshops: (1) A good scheduling method can not only improve customer satisfaction and reduce production costs, but also reduce the interference of human factors through replacing some of the functions of the workshop manager. (2) The stability and excellent scheduling results of the adaptive scheduling strategy proves that the proposed method is an effective method to realize an intelligent unmanned workshop.

## 7. Conclusions

This paper focuses on the enterprise’s manufacturing decisions by considering a high disturbance environment. A real-time status model of heterogeneous resources in SSF is established. The mathematical model is formulated with three objectives, i.e., the makespan, energy consumption, and mean tardiness of jobs. A real-time scheduling algorithm based on the information entropy theory (RTSIET) based on the dynamic service capability of manufacturing resources is proposed. Then, a real-world case is employed to explore the scheduling method in a high disturbance and multi-resource environment. In addition, the algorithm performance under random arrival jobs is analyzed. The proposed model is solved by RTSIET, which can obtain the real-time multi-objective weight value by calculating the standard information entropy of each job. The agility and stability of the system are demonstrated through a practical application.

We will apply this real-time scheduling strategy to a combination of other methods in our future research. The game theory may be a good candidate due to its multi-resource collaboration ability.

## Figures and Tables

**Figure 1 sensors-20-04507-f001:**
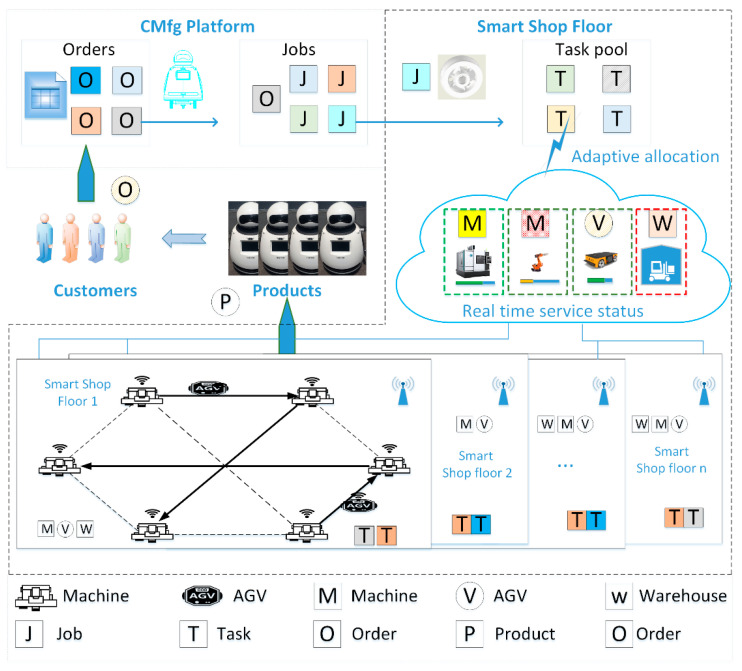
Scenario description of the real-time manufacturing resources allocation (RTMSA) problem.

**Figure 2 sensors-20-04507-f002:**
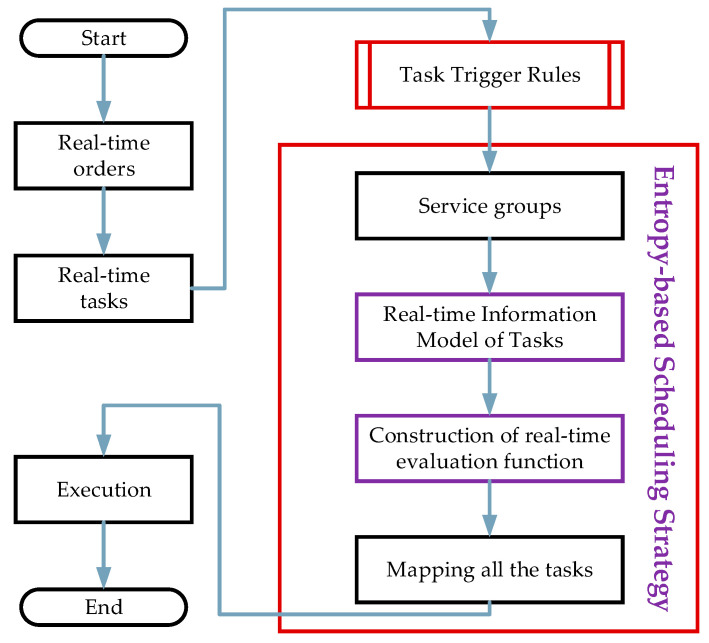
The flowchart of production and logistics resources (PLRs) adaptive scheduling.

**Figure 3 sensors-20-04507-f003:**
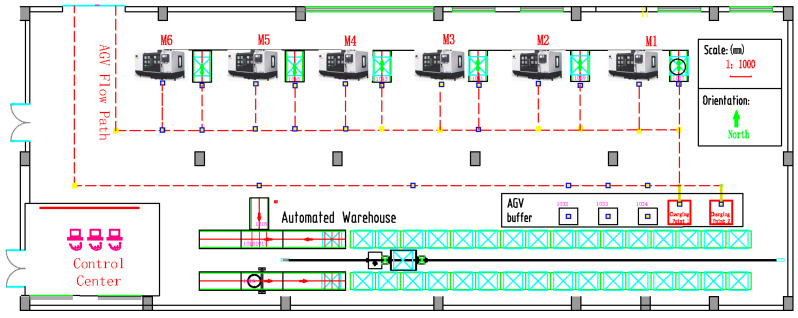
The layout of the hub workshop.

**Figure 4 sensors-20-04507-f004:**
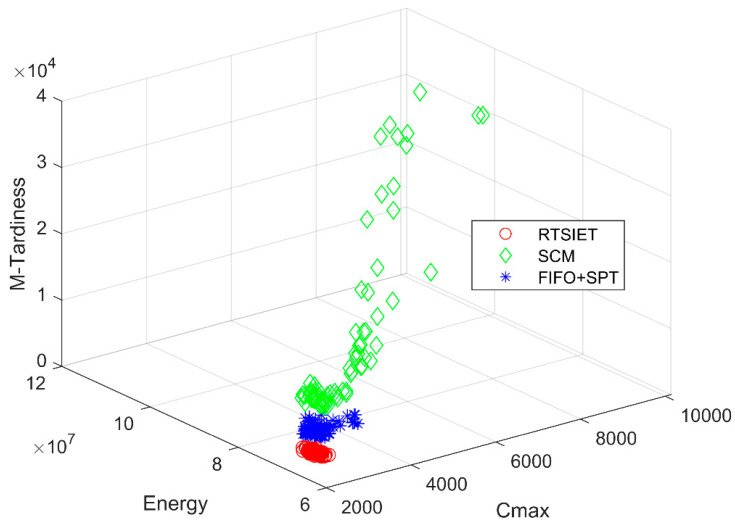
Pareto front of three algorithms in solving the proposed problem.

**Table 1 sensors-20-04507-t001:** The notation used in the study.

Notations	Description
$job_{set}$	Job set
$job^{k}$	*k-*th job
$tk_{n}^{k}$	n-th operation of job k, $job^{k}=\left(tk_{n}^{k}|n=1,2,\cdots ,l\right)$
$s_{j}^{t}$	Type of service that a machine can provide
$pp_{nj}^{k}$	Power of $m_{j}$ for the operation of $tk_{n}^{k}$
$p_{nj}^{k}$	Service time of $m_{j}$ for $tk_{n}^{k}$
$pp_{ni}^{k}$	Power of the AGV i for the operation of $tk_{n}^{k}$
$p_{ni}^{k}$	Service time of the AGV $a_{i}$ for $tk_{n}^{k}$
$bp_{nj}^{k}$	The time when the machine starts to the operation of $tk_{n}^{k}$
$fp_{nj}^{k}$	Completion time for $tk_{n}^{k}$
$bp_{ni}^{k}$	Start time of $a_{i}$ to operate $tk_{n}^{k}$
$mp_{ni}^{k}$	The time AGV arrives at the machine where the $tk_{n}^{k}$ is located
$fp_{ni}^{k}$	The time of $a_{i}$ completes the $tk_{n}^{k}$
$p_{j}^{I}$	Idle power of $m_{j}$
$\bar{v}$	Speed of an AGV
$L_{i}$	Location of $a_{i}$
${\bar{m}}_{j}^{t}$	Capacity of $m_{j}$ at time *t*
${\bar{a}}_{i}^{t}$	Handling capacity of $a_{i}$ at time *t*
$\bar{M}$	Optional machine set
$\bar{A}$	Optional AGV set
$Tk^{t}$	A set of tasks in the task-pool
$sq_{j}$	Service queue of $m_{j}$
$I_{j}$	Total idle time of $m_{j}$
$sq_{i}$	Service queue of AGV $a_{i}$
$C^{k}$	Completion time of $job^{k}$
$d^{k}$	Due date for $job^{k}$
$L^{k}$	Lateness of $job^{k}$

**Table 2 sensors-20-04507-t002:** The distance among the warehouse and machines. ($m_{0}$ is warehouse, $m_{1}\sim m_{6}$ are machines).

Distance [m]	*m* _0_	*m* _1_	*m* _2_	*m* _3_	*m* _4_	*m* _5_	*m* _6_
*m* _0_	0	40	46	52	60	66	75
*m* _1_	40	0	6	12	16	24	33
*m* _2_	46	6	0	12	18	24	33
*m* _3_	52	12	6	0	6	12	21
*m* _4_	60	18	12	6	0	6	15
*m* _5_	66	24	18	12	6	0	9
*m* _6_	75	33	27	21	15	9	0

**Table 3 sensors-20-04507-t003:** The distance between the warehouse and each machine.

Time [s]\ Power [kW/h]		*m* _0_	*m* _1_	*m* _2_	*m* _3_	*m* _4_	*m* _5_	*m* _6_
*job*	1CT	180\3.74	190\3.11	170\4.38	180\4.24	190\3.41	200\4.5	180\3.74
	2TU	170\4.38	190\4.11	170\4.48	170\4.59	180\4.24	200\3.95	170\4.38
	3GR	170\4.06	190\3.18	170\3.70	170\4.08	180\5.82	200\4.08	170\4.06
	4DR	230\4.18	240\4.13	250\3.20	230\4.19	240\4.09	200\5.01	230\4.18
	5TA	220\5.40	220\5.39	240\4.17	230\5.28	240\4.68	260\4.57	220\5.40

**Table 4 sensors-20-04507-t004:** Idle power of machines.

*m* _i_	*m* _0_	*m* _1_	*m* _2_	*m* _3_	*m* _4_	*m* _5_	*m* _6_
Idle Power [kW/h]	0.98	1.23	1.48	1.06	1.06	1.16	1.27

**Table 5 sensors-20-04507-t005:** Power\speed of automated guided vehicle (AGVs).

AGV	*a_i_*
Power [kW/h]	1
Speed [m/s]	0.5

**Table 6 sensors-20-04507-t006:** Results comparison with the real-time scheduling algorithm based on the information entropy theory (RTSIET), self-adaptive collaboration method (SCM), first in first out (FIFO) + longest processing time (LPT), and FIFO + the shortest processing time (SPT).

NA	RTSIET						SCM					
	$C_{max}\;[s]$		E [10,000 J]		$\bar{T}$ [s]		$C_{max}\;[s]$		E [10,000 J]		$\bar{T}$ [s]	
	MV	V	MV	V	MV	V	MV	V	MV	V	MV	V
1	3898	179	7857	101	108	5	5881	2893	8763	1558	1932	143
2	3135	55	7416	102	0	0	5328	2661	8195	1259	1518	118
3	3040	72	7281	95	0	0	5266	2741	8089	1314	1346	122
4	2989	53	7204	101	0	0	5197	2723	8014	1328	1234	124
5	2975	51	7165	90	0	0	5202	2695	7978	1221	1004	117
**NA**	**FIFO + LPT**						**FIFO + SPT**					
	$C_{max}\;$ **[100 s]**		**E [100,000 J]**		$\bar{T}\;$ **[100 s]**		$C_{max}$ **[s]**		**E [10,000 J]**		$\bar{T}$ **[s]**	
	**MV**	**V**	**MV**	**V**	**MV**	**V**	**MV**	**V**	**MV**	**V**	**MV**	**V**
1	193	4.20	1840	37.5	197	6.37	3918	157	8201	127	1149	927
2	192	4.17	1828	36.4	196	6.36	3559	82	7873	121	965	411
3	190	3.56	1812	31.1	193	5.54	3498	71	7777	115	784	351
4	187	3.03	1787	27.8	190	4.88	3509	73	7769	110	965	363
5	188	2.94	1736	34.4	181	3.69	3639	79	7761	991	857	371

Notes: NA represents the number of AGVs; E represents total energy consumption; $\bar{T}$ represents mean tardiness; $C_{max}$ represents makespan; MV represents mean value; V represents variance.

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
