# Peer review of "Real-Time Production and Logistics Self-Adaption Scheduling Based on Information Entropy Theory"

_sensors, 2020, doi:10.3390/s20164507_

Round 1

Reviewer 1 Report

The paper “Real-time Production and Logistics Self-adaption Scheduling based on Information Entropy Theory” is very interesting and the topic is adequate for the journal. The paper develops real-time model of production and logistics resources, designs a task entropy model based on the task information, and realizes a real-time self-adaption collaboration of production and logistics resources. The research was carried out with a proper methodology, the results are original.

As for the research framework, the conceptual background and the research aim are very clear. As for methodology, an in-depth description of the flowchart summarizing the proposed algorithm and the different methodological steps will help the readers to understand the methodology. In addition, the authors should describe and justify more in details the choice of how their application converge or diverge from other similar studies provided in the literature. My suggestion is to compare the results obtained with your model with other future studies that could be based on genetic or swarm-based algorithm. To make this comparison and identify the possibility to adopt an alternative methodological approach for job shop scheduling problem with random job arrival time in the future research avenues of the study, you may refer to Bottani et al. (2018) (Solving machine loading problem of flexible manufacturing systems using a modified discrete firefly algorithm. International Journal of Industrial Engineering Computations, 8(3), 363-372) and Todd et al. (2018) (Firefly-Inspired Algorithm for Job Shop Scheduling), in which the authors used a discrete firefly algorithm to solve one of the most common multicriteria decision making problem. With regard to the main results, could the context of investigation affect the results? Why? Please stress managerial implications of the study.

Author Response

Dear reviewer,

We submit our manuscript entitled “time Production and Logistics Self-adaption Scheduling based on Information Entropy Theory” (ID: sensors-866659).

This paper belongs to the Special Issue of “Internet of Things”. In addition, this paper belongs to the “Internet of Things, Big Data and Smart Systems”.

We would like to thank you for your highly constructive comments that have helped us to improve this paper greatly.

We have carefully modified this manuscript according to your comments. Our responses to each comment and the revised manuscript are attached.

We appreciate your consideration of our manuscript.  

Best regards, 

Wenchao Yang (on behalf of all authors)

School of Logistics Engineering,

Wuhan University of Technology,

Wuhan 430063, China.

Email: yangwc6532@gmail.com

Reviewer 2 Report

The paper discusses method of Real-Time Scheduling algorithm based on Information Entropy Theory. From my point of view, the article is very well written, the chapters are logically chosen and follow each other smoothly. Firstly, the real-time model of production and logistics resources is described. Then, the task entropy model is built based on the task information. Finally, the real-time self-adaption collaboration of production and logistics resources is realized. Based on a practical case study, the proposed algorithm is carried out to evaluate its effectiveness.

I recommend to accept the article as it is.

Author Response

Dear Reviewer,

Thank you for your reviewer’s comments concerning our manuscript entitled “time Production and Logistics Self-adaption Scheduling based on Information Entropy Theory” (ID: sensors-866659).

We would like to thank you for your recognition of our research work.

Best Wenchao Yang

School of Logistics Engineering,

Wuhan University of Technology,

Wuhan 430063, China,

Email: yangwc6532@gmail.com

Reviewer 3 Report

The authors of the paper illustrate the enterprise’s manufacturing decisions by considering high disturbance environment. The subject of the paper is good and fits with the purpose of the journal. However, the style of the paper has to be improved.

The novelty of the paper should be highlighted more clearly in all sections.

Figure 2 is very poor. It doesn't look professional. The authors have to include scale, orientation symbol and locator.

All tables should be on the same page, do not break any table in the manuscript (such as, Table 1 and 2).

I could not see any related works to Pareto front in the text.

The introduction section is weak, the motivation, justification, and contribution should be more emphasizes in the introduction section. Moreover, it needs presenting more related works about improving production and logistics real-time.

- The structure of the manuscript should be based on below sections:

1- Introduction
1.1. motivation and contribution of the study
2. Related works or literature review
3. Methodology
4. Case study

5.Results
6. Discussion
7. Conclusion

References

Author Response

(The authors gave the same response as above.)

Round 2

Reviewer 1 Report

The paper was improved according to the review comments.

Author Response

Thanks for your recognition of our research work.

Reviewer 3 Report

The authors improved the paper according  to the suggested points.

The font has to be modified considering the journal’s format style.

I recommend citing the following work for Line 124:

https://doi.org/10.1016/j.eswa.2018.08.049

Author Response

Thanks for your kind suggestion. We have now cited more references into our revised manuscript. The corresponding references is [33] for Line 131.